# Freshwater *Chlorobia* Exhibit Metabolic Specialization among Cosmopolitan and Endemic Populations

Sarahi L. Garcia,[a,b] Maliheh Mehrshad,[a,c] Moritz Buck,[a,c] Jackson M. Tsuji,[d*] Josh D. Neufeld,[d] Katherine D. McMahon,[e,f] Stefan Bertilsson,[a,c] Chris Greening,[g] Sari Peura[h]

[a]Department of Ecology and Genetics, Limnology, Uppsala University, Uppsala, Sweden

[b]Department of Ecology, Environment, and Plant Sciences, Science for Life Laboratory, Stockholm University, Stockholm, Sweden

[c]Department of Aquatic Sciences and Assessment, Swedish University of Agricultural Sciences, Uppsala, Uppsala, Sweden

[d]Department of Biology, University of Waterloo, Waterloo, Ontario, Canada

[e]Department of Civil and Environmental Engineering, University of Wisconsin, Madison, Madison, Wisconsin, USA

[f]Department of Bacteriology, University of Wisconsin, Madison, Madison, Wisconsin, USA

[g]Department of Microbiology, Biomedicine Discovery Institute, Monash University, Clayton, Victoria, Australia

[h]Department of Forest Mycology and Plant Pathology, Science for Life Laboratory, Swedish University of Agricultural Sciences, Uppsala, Uppsala, Sweden

Sarahi L. Garcia, Maliheh Mehrshad, and Moritz Buck contributed equally to this article. The order of the authors was designated by time commitment in the writing process.

**ABSTRACT** Photosynthetic bacteria from the class *Chlorobia* (formerly phylum *Chlorobi*) sustain carbon fixation in anoxic water columns. They harvest light at extremely low intensities and use various inorganic electron donors to fix carbon dioxide into biomass. Until now, most information on the functional ecology and local adaptations of *Chlorobia* members came from isolates and merely 26 sequenced genomes that may not adequately represent natural populations. To address these limitations, we analyzed global metagenomes to profile planktonic *Chlorobia* cells from the oxyclines of 42 freshwater bodies, spanning subarctic to tropical regions and encompassing all four seasons. We assembled and compiled over 500 genomes, including metagenome-assembled genomes (MAGs), single-amplified genomes (SAGs), and reference genomes from cultures, clustering them into 71 metagenomic operational taxonomic units (mOTUs or "species"). Of the 71 mOTUs, 57 were classified within the genus *Chlorobium*, and these mOTUs represented up to ~60% of the microbial communities in the sampled anoxic waters. Several *Chlorobium*-associated mOTUs were globally distributed, whereas others were endemic to individual lakes. Although most clades encoded the ability to oxidize hydrogen, many lacked genes for the oxidation of specific sulfur and iron substrates. Surprisingly, one globally distributed Scandinavian clade encoded the ability to oxidize hydrogen, sulfur, and iron, suggesting that metabolic versatility facilitated such widespread colonization. Overall, these findings provide new insight into the biogeography of the *Chlorobia* and the metabolic traits that facilitate niche specialization within lake ecosystems.

**IMPORTANCE** The reconstruction of genomes from metagenomes has helped explore the ecology and evolution of environmental microbiota. We applied this approach to 274 metagenomes collected from diverse freshwater habitats that spanned oxic and anoxic zones, sampling seasons, and latitudes. We demonstrate widespread and abundant distributions of planktonic *Chlorobia*-associated bacteria in hypolimnetic waters of stratified freshwater ecosystems and show they vary in their capacities to use different electron donors. Having photoautotrophic potential, these *Chlorobia* members could serve as carbon sources that support metalimnetic and hypolimnetic food webs.

**KEYWORDS** *Chlorobia*, freshwater, photosynthetic bacteria, planktonic

Address correspondence to Sarahi L. Garcia, sarahi.garcia@su.se.

*Present address: Jackson M. Tsuji, Institute of Low Temperature Science, Hokkaido University, Sapporo, Japan.

Although oxygenic phototrophs dominate contemporary carbon fixation, anoxygenic phototrophs have been important over planetary time scales and continue to occupy important niches in a wide range of ecosystems (1). Among the anoxygenic phototrophs, known members of the class *Chlorobia* (green sulfur bacteria, formerly phylum *Chlorobi*) are photolithoautotrophic anaerobes (2, 3) that leverage bacteriochlorophyll-rich organelles called chlorosomes to harvest light at extremely low intensities (i.e., in the range of 1 to 10 nmol photons $m^{-2}$ $s^{-1}$) (4). Such efficient light harvesting determines the ecological niche of these bacteria at the lowermost strata of the photic zone in stratified water columns, where the least amount of light is available (5, 6). Studies of isolates obtained from aquatic systems and microbial mats have revealed that members of the class *Chlorobia* use electrons derived from reduced sulfur compounds and/or hydrogen to fix carbon dioxide via the reverse tricarboxylic acid (rTCA) cycle (7–9). Although several strains can oxidize thiosulfate, they almost universally use sulfide as an electron donor for $CO_2$ reduction and source of sulfur for assimilation (2). Several members of the class *Chlorobia* are additionally known to oxidize ferrous iron ($Fe^{2+}$) in a process called photoferrotrophy (10). Such photoferrotrophs include *Chlorobium ferrooxidans*, which obtains sulfur via assimilatory sulfate reduction, along with three other recently characterized strains (11–14). Although all characterized *Chlorobia* pure cultures can grow with carbon dioxide ($CO_2$) or bicarbonate as their sole carbon source, many isolates have been reported to assimilate simple organic acids, such as acetate and pyruvate, under photomixotrophic or photoheterotrophic conditions (2). Moreover, most *Chlorobia* cells also encode the potential for nitrogen fixation (15–17). Together, this metabolic capacity and versatility enable members of the class *Chlorobia* to colonize dimly lit and anoxic aquatic environments.

In stratified freshwater ecosystems, *Chlorobia* microorganisms have particularly important ecological and biogeochemical roles (18–21). They accumulate to form dense populations at certain strata, manifesting as deep chlorophyll maxima (22), and contribute to carbon budgets of such lakes (23, 24), while concomitantly contributing to sulfur, iron, hydrogen, and/or nitrogen cycling (17, 25, 26). Despite progress, knowledge of the geographical distribution and metabolic capabilities of *Chlorobia* members within natural planktonic communities remains incomplete. Most understanding of the metabolic capabilities and ecophysiological strategies of this class come from studies of approximately 100 isolates (5), with genomes available for only 26 of these representatives (2, 27, 28). Especially given that cultures are often poor representatives of natural populations, the diversity and distribution of planktonic *Chlorobia* members are largely unknown, and almost nothing is understood about how genes conferring distinct ecophysiological traits are distributed among *Chlorobia*-associated genomes from geographically distinct water bodies.

Here, we used a cultivation-independent approach to gain a more comprehensive understanding of the natural distributions and metabolic capabilities of *Chlorobia* members within 42 freshwater bodies that are distributed across boreal, subarctic, and tropical regions in Europe and North America (Fig. 1). We assembled and analyzed over 500 genomes of *Chlorobia* members, including 465 metagenome-assembled genomes (MAGs) and 19 single-amplified genomes (SAGs). These were further clustered into 71 metagenome operational taxonomic units (mOTUs). Each mOTU is a cluster of genomes that share 95% average nucleotide identity (ANI). We observed that some of these mOTUs exhibited cosmopolitan distributions, whereas others were locally constrained and possibly endemic to specific lakes or regions. In addition, clades varied in their encoded metabolic capabilities, and we found several mOTUs that share substrate specialization profiles. These findings reveal further metabolic versatility and niche specialization within abundant *Chlorobium*-associated bacteria.

## RESULTS AND DISCUSSION

**Metagenome assembly and characterization reveals that most planktonic members of the class *Chlorobia* affiliate with the genus *Chlorobium*.** We investigated the planktonic *Chlorobia*-affiliated populations of 30 lakes and 12 ponds

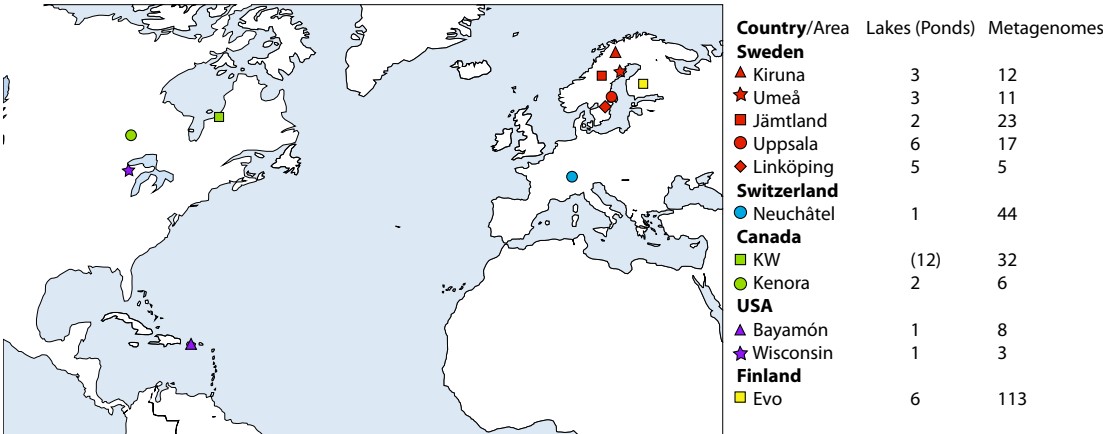

| Country/Area | Lakes (Ponds) | Metagenomes |
|---|---|---|
| **Sweden** | | |
| ▲ Kiruna | 3 | 12 |
| ★ Umeå | 3 | 11 |
| ■ Jämtland | 2 | 23 |
| ● Uppsala | 6 | 17 |
| ◆ Linköping | 5 | 5 |
| **Switzerland** | | |
| ● Neuchâtel | 1 | 44 |
| **Canada** | | |
| ■ KW | (12) | 32 |
| ● Kenora | 2 | 6 |
| **USA** | | |
| ▲ Bayamón | 1 | 8 |
| ★ Wisconsin | 1 | 3 |
| **Finland** | | |
| ■ Evo | 6 | 113 |

**FIG 1** Map illustrating the lake and pond metagenome sampling sites for mapping and recovery of metagenome-assembled genomes (MAGs) and single-amplified genomes (SAGs). Symbols are colored by country, and a unique shape is used for each area within the same country. "KW" represents Kujjuarapik-Whapmagoostui.

sampled from Europe and North America (Fig. 1). Most of these samples were collected with the objective of investigating depth-associated changes in taxonomic diversity in different water masses that capture oxic epilimnion, metalimnion, and anoxic hypolimnion conditions (29). Following assembly and binning of individual metagenomes, we compiled 454 new MAGs that affiliated with the class *Chlorobia*. Moreover, we collected 19 SAGs belonging to the class *Chlorobia* from two of the sampled lakes. To characterize the phylogenetic distributions of these new MAGs and SAGs, we further compiled 25 genomes available from the Genome Taxonomy Database (GTDB) and 11 MAGs available from previous studies (27, 30–32). In total, the data set included 509 genomes, including the MAGs, SAGs, and complete genomes of isolates (see Table S1 in the supplemental material). Genome completeness varied from 50% to 100%, with an average of 89.0% and a median of 94.5% (Table S1); contamination was below 5% in all cases.

We clustered recovered genomes by 95% average nucleotide identity (ANI), which has previously been shown to unite classical species definitions and separate sequenced strains into consistent and distinct groups (33–35). Clustering into mOTUs provided a more comprehensive understanding of the genomic structure of genus *Chlorobium* members. We obtained 71 metagenomic operational taxonomic units (mOTUs) belonging to the class *Chlorobia* (see Table S2 in the supplemental material), of which 57 mOTUs were classified as members of the genus *Chlorobium* based on GTDB taxonomy (Fig. 2; Table S2) (27). These 57 mOTU genomes had an average completeness of approximately 90% and an estimated genome size of 2.6 Mbp (range, 2.1 to 3.7 Mbp), which are consistent with previous studies showing that *Chlorobium* genomes from isolates range from 1.9 to 3.3 Mbp (2). Among 57 *Chlorobium*-associated mOTUs, 13 were composed exclusively of different genomes from previously described nonidentical isolates (2), including *Chl. phaeobacteroides*, *Chl. limicola*, *Chl. luteolum*, *Chl. ferrooxidans*, and *Chl. phaeoclathratiforme* (Fig. 2; and Table S2). None of the mOTUs included both isolate genomes and environmental MAGs. Whereas several of the cultured members of the class *Chlorobia* were isolated from lakes (36), our analysis suggests that the majority of natural diversity within freshwater planktonic members of the class *Chlorobia* is represented by uncultivated members of the genus *Chlorobium*.

**The genus *Chlorobium* includes both cosmopolitan and endemic lineages.** By identifying MAGs from different source locations that clustered within the same mOTUs, we were able to determine whether a "species" is widely distributed and putatively cosmopolitan or endemic to specific locations. To investigate the abundance

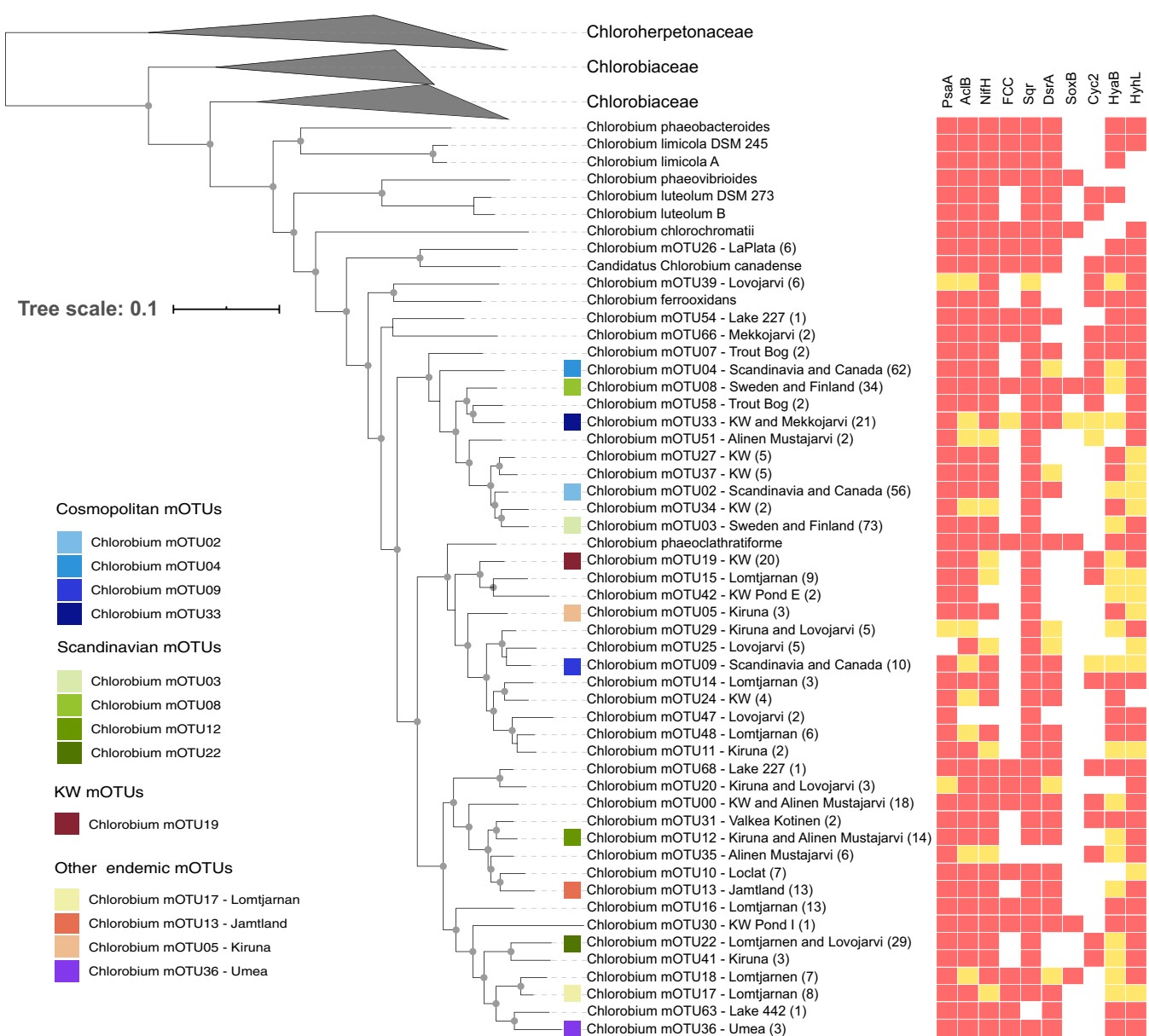

**FIG 2** Phylogenetic relationships, metabolic capabilities, and ecological distributions of *Chlorobium* mOTUs. The tree was constructed using GTDB-Tk "de-novo" alignment and was annotated and curated in iTOL. Isolates are shown in the tree with species names instead of mOTU numbers. Other tips on the tree show the genus name and mOTU number followed by the location from which the genomes were assembled. If the MAGs were assembled from more than one lake in the same area, then the name of the area is written. If the MAGs were assembled from more than one area in the same country, then the name of the country is written. Lake La Plata is in Bayamon, Puerto Rico; Lakes 227 and 442 are in the IISD-ELA (near Kenora, Canada). Lake Lovojarvi, Lake Mekkojarvi, Lake Valkea Kotinen, and Lake Alinen Mustajarvi are in Evo, Finland. Lake Trout Bog is in Wisconsin, Lake Loclat is in Neuchatel, Switzerland, and Lake Lomtjarnen is in Jamtland, Sweden. Numbers in parentheses show the number of genomes in each mOTU. The genus *Chlorobium* is part of the family *Chlorobiaceae*. Other genomes in different genera within the family *Chlorobiaceae* are clustered. The tree also includes information about the presence/absence of several genes, as indicated by their products: i.e., PsaA (photosystem I P700 chlorophyll *a* apoprotein A1), AclB (ATP-citrate lyase beta-subunit), NifH (nitrogenase iron protein), FCC (flavocytochrome *c* sulfide dehydrogenase), Sqr (sulfide-quinone oxidoreductase), DsrA (reverse dissimilatory sulfite reductase), SoxB (thiosulfohydrolase), Cyc2 (iron-oxidizing outer membrane *c*-type cytochrome), HyaB (group 1d [NiFe]-hydrogenase large subunit), and HyhL (group 3b [NiFe]-hydrogenase large subunit). In the heat map, red indicates that the gene is present in the core genome of the corresponding mOTU, and yellow indicates that it is present in an accessory genome of the mOTU. White indicates that the gene is absent in the corresponding mOTU. The 13 most abundant *Chlorobium* mOTUs in the investigated environments have a colored square before the name of the mOTU. Gray circles represent bootstrap values higher than 50%. Please see Table S2 for information on all 71 mOTUs.

and prevalence of each reconstructed mOTU within resident *Chlorobia* assemblages, we mapped metagenomic reads from the lake and pond data sets (see Table S3 in the supplemental material) against the genome collection (Table S1). Reads of our lake/pond data set mapped to 45 (out of 71) *Chlorobia* mOTUs, of which 42 mOTUs were classified as members of the genus *Chlorobium* (Fig. 3). None of these mOTUs

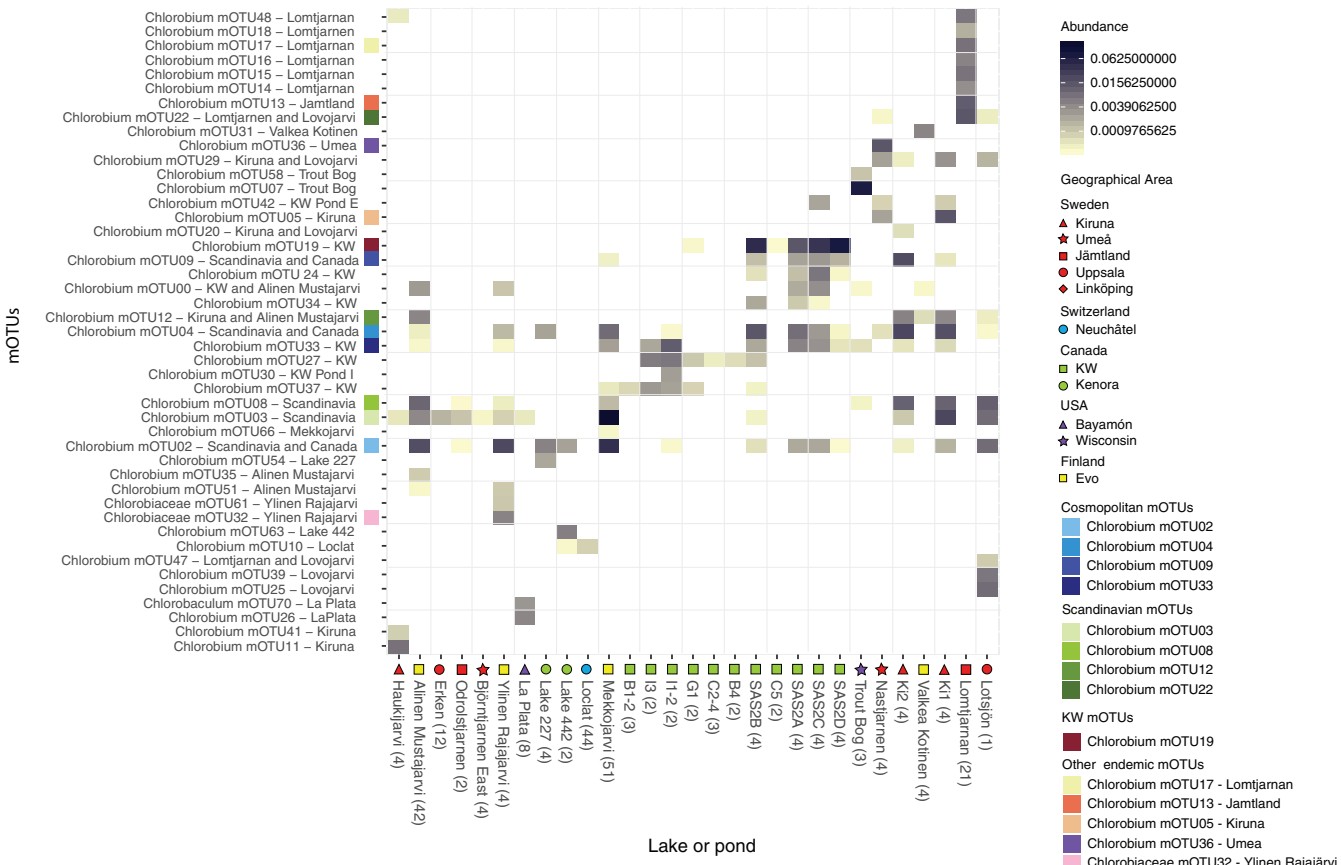

**FIG 3** Relative abundance of metagenome reads for the 45 *Chlorobia* mOTUs present in lake and pond metagenomes. The top 14 *Chlorobia* mOTUs are color coded according to the location from which they were assembled. Metagenomes were subsampled to one million reads and then mapped competitively with 100% identity cutoff to all genomes in all mOTUs. The reads were normalized to the relative abundance of reads per metagenome. Relative abundances from all depths and time points were averaged, and the number of the samples averaged is shown in the parentheses next to the lake or pond name. The cutoff for presence of an mOTU to be included was 0.03% (i.e., 0.0003 in the figure) read abundance per lake/pond. The name of the mOTUs includes geographical information about the origin of the genomes. The lakes or ponds shown along the *x* axis include a symbol for the region where they are located.

corresponded to cultivated *Chlorobium* representatives. This provides further evidence that uncultivated *Chlorobium* members dominated planktonic *Chlorobia* present in sampled lakes and ponds.

Out of the 42 locations represented in this study, we found that metagenomes sequenced from 18 lakes (of 30 total) and 11 ponds (of 12 total) contained *Chlorobia*-affiliated reads (Fig. 3). However, for 10 of the lakes that did not show *Chlorobia* read abundance, we only have epilimnion samples with full oxygenation, and so it was expected to not find *Chlorobia* reads or genomes. We found four mOTUs composed of MAGs reconstructed from assemblies originating from both studied continents, i.e., mOTU02, mOTU04, mOTU09, and mOTU33 (in blue in Fig. 2 to 5), and we define these mOTUs as cosmopolitan because of their broad distributions. Metagenomic read recruitment confirmed that these cosmopolitan mOTUs were present in lakes and ponds from both studied continents (Fig. 3). In contrast, the other 41 mOTUs were less widely distributed and were found in just one lake, in one geographical area, or just in Scandinavia. The apparent cosmopolitan or locally constrained clades did not appear to be monophyletic, but rather distributed across the phylogenetic tree of the class *Chlorobia* (Fig. 2).

Depth stratification of *Chlorobia* mOTUs was evident when mapped reads were visualized as relative abundances, based on total metagenomic reads, across lake water column samples (Fig. 4). As expected, our study showed that *Chlorobium* mOTUs were found mostly below the oxycline (Fig. 4), and several were highly abundant within their

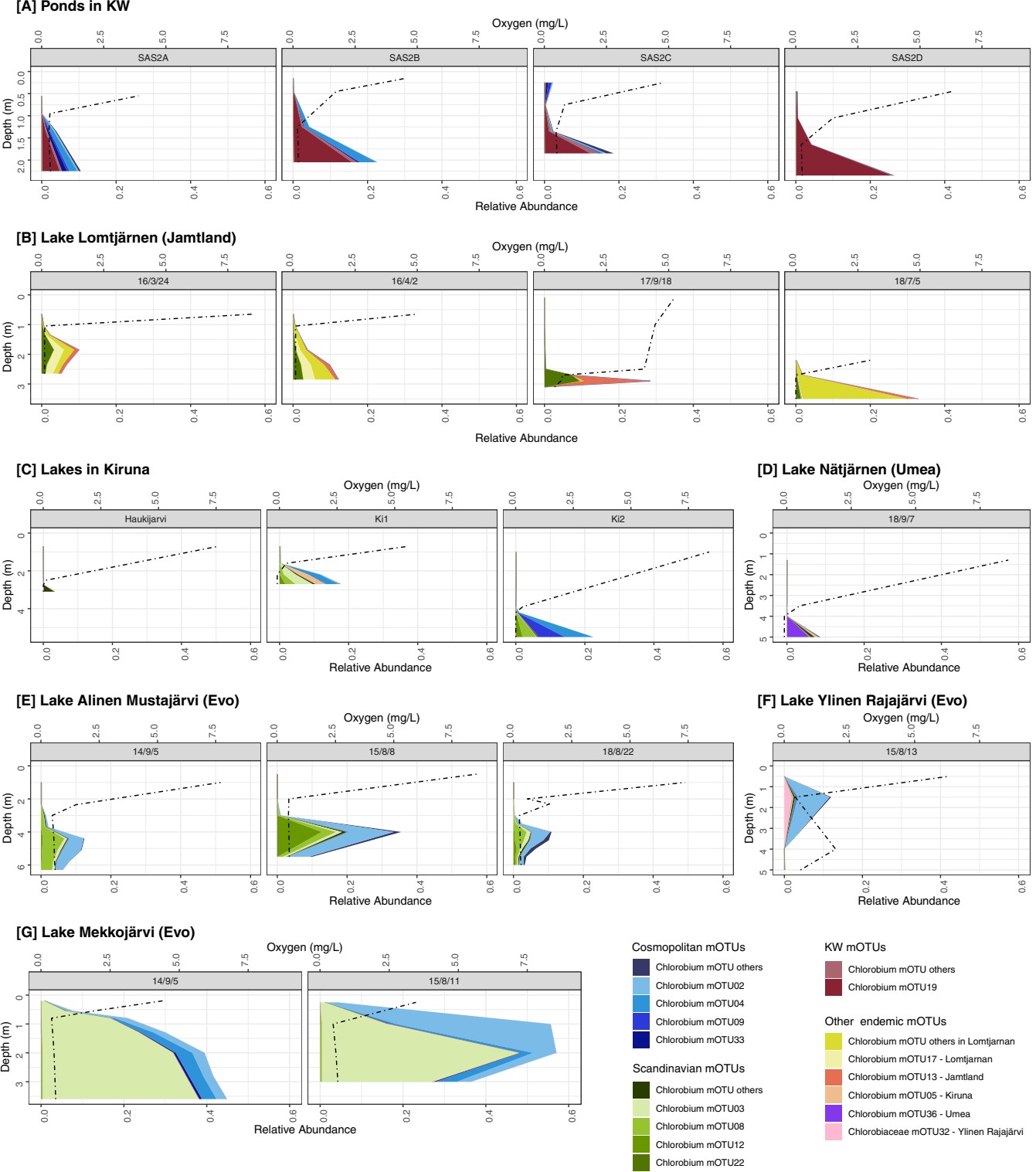

**FIG 4** Distribution of environmentally abundant mOTUs across depth profiles in the lake and pond data sets with depth-discrete sampling. Note that the 14 most abundant mOTUs have individual color coding, and the remaining abundant mOTUs are combined in several categories labeled "other." Oxygen is represented by a dashed line. (A) Ponds in the KW area. (B) Time points for Lake Lomtjärnen in Sweden. (C) Different lakes in the Kiruna area in Sweden. (D) Lake Nästjärnen in Sweden. (E) Time points of Lake Alinen Mustajärvi in Finland. (F) Lake Ylinen Rajajärvi in Finland. (G) Time points for Lake Mekkojärvi in Finland.

respective aquatic microbial communities. In particular, *Chlorobium* sequences comprised 57% of the community at several depths of Lake Mekkojärvi, Finland (Fig. 4G). Such relative abundances are consistent with previous studies showing that this class can constitute 12 to 47% of reads from lake ecosystems (31, 32, 37, 38). Our data show one clear example of multiple *Chlorobium* populations typically coinhabiting the same lake with spatially separated niches. Specifically, mOTU02 and mOTU03 were differentially distributed by depth in August 2015 within Lake Mekkojärvi samples (Fig. 4G). Similar findings were previously reported in Trout Bog, WI, where one of the populations was recovered only in the lowermost water layers (i.e., mOTU07, GSB-A/ Chlorobium-111), whereas the other populations had broader distributions (e.g., mOTU58, GSB-B) (30, 32). Previous research suggests that such niche specialization may be explained by distinct pigment absorbance profiles (32, 39).

Several mOTUs appear to be endemic to specific geographical and seasonal niches. For example, we observed that mOTUs 22, 17, and 13 were temporally stable within Lake Lomtjärnen, Sweden (Fig. 4B). These mOTUs were observed across four time points: March 2016 (under ice and snow cover), April 2016 (ice cover only), September 2017 (ice free), and July 2018 (ice free). The oxycline is lower in the water column when the lake is ice free, and *Chlorobium* populations were abundant below the oxycline at all time points. The mOTUs residing in this lake were mostly endemic to the lake or to the region, and they were consistently recovered at different times. In addition, the relative abundance of these mOTUs changed according to the sampling time. For example, mOTU13 increased in relative abundance at the September sampling point. We predict that some endemic mOTUs may occupy specific seasonal niches and suggest that a more temporally resolved sampling effort, together with cell counts to calculate absolute abundances, would help test this hypothesis.

**The metabolic potential of the most abundant *Chlorobium* groups.** We investigated the metabolic potential of all *Chlorobium*-associated mOTUs. Combining ANI clustering and statistical approaches allowed us to calculate the probability that specific genes are present in the core or accessory genomes of the "species," despite the incompleteness of several MAGs and SAGs. Overall, 42 out of the 71 *Chlorobia* mOTUs were composed of more than one genome, and hence we found power in replication (40). For example, mOTU08 included 34 MAGs with an average completeness of 95%, and the mOTU contained all surveyed functional genes (Fig. 2) in the core genome, except for HyaB, which was putatively within the accessory genome. Moreover, in order for a gene to be completely missing from the mOTU, the gene had to be absent in all of the MAGs and SAGs associated with that particular mOTU. The absence of a gene in our mOTU data is strong evidence for the absence of that gene from the corresponding population, unlike traditional MAG-based studies that select a single representative MAG for analysis.

We found an average of 710 annotated genes with designated KEGG functions per mOTU. For these mOTUs, we were able to investigate the core metabolic functions of freshwater members of the genus *Chlorobium*. As expected, given the photolithoautotrophic lifestyle of cultured *Chlorobia*, core features of the genomes included all genes related to glycolysis and gluconeogenesis, reverse TCA cycle, and chlorophyll and bacteriochlorophyll biosynthesis (Fig. 2 and 5; see Fig. S1A in the supplemental material). These findings are consistent with previous studies showing very minor differences in genes encoding the photosynthetic type I reaction center unit in this class (41). Molybdenum- and iron-nitrogenases were also widely encoded by the representatives of the class *Chlorobia* (Fig. S1C). Thus, light harvesting, carbon fixation, and nitrogen fixation appear to be conserved core metabolic features of freshwater *Chlorobia* members.

In contrast to functions that were widely encoded, the products of *Chlorobia*-associated mOTUs varied significantly in their encoded capacity to use electron donors (i.e., hydrogen, sulfide, thiosulfate, and iron). With respect to sulfur compounds, most mOTUs were capable of oxidizing sulfide via one or more enzymes. Sulfide-quinone

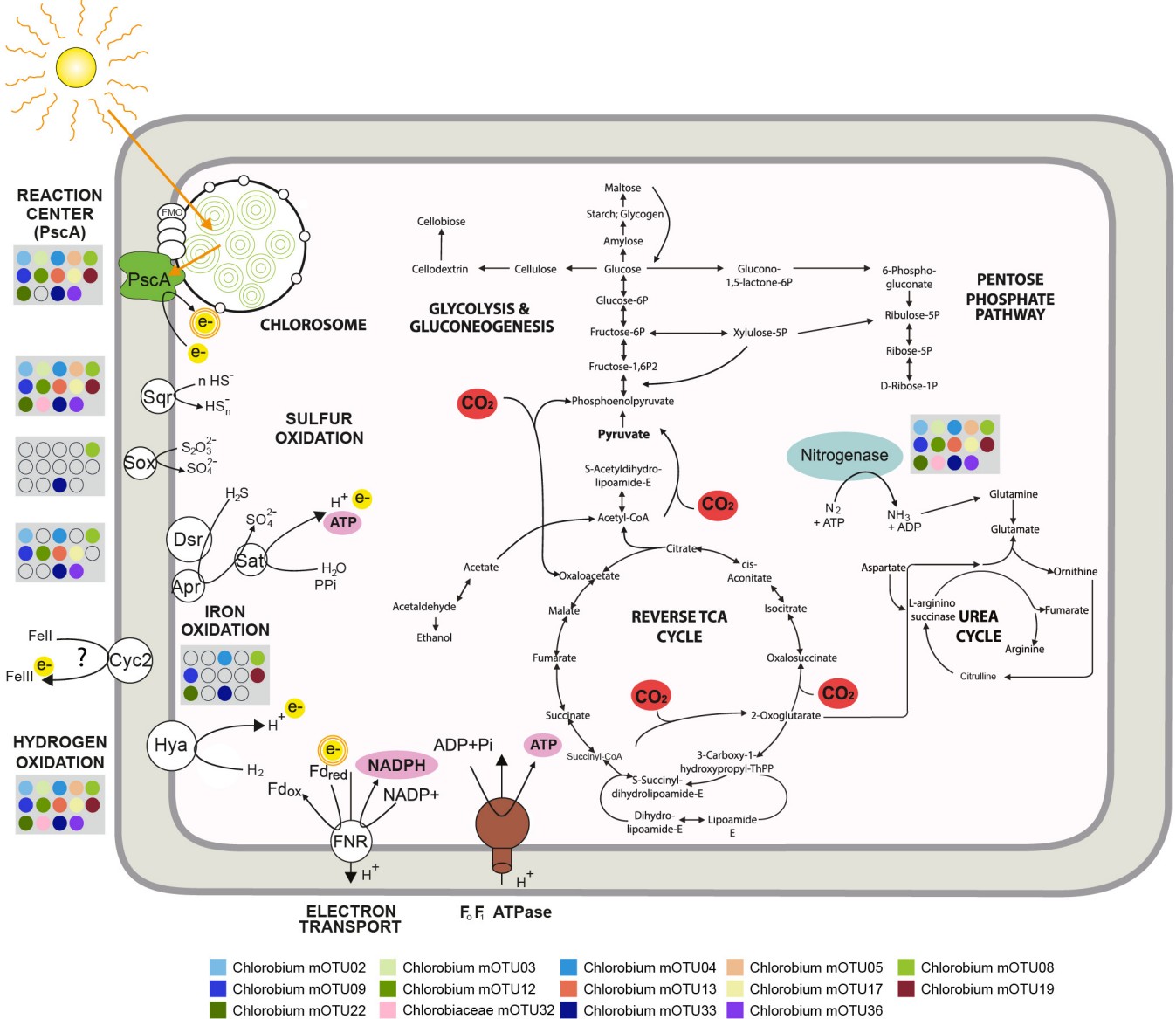

**FIG 5** Metabolic potential of the 14 most abundant *Chlorobia* mOTUs detected in the studied freshwater ecosystems, with a focus on photosynthesis, electron transport, and carbon fixation. Blue indicates nitrogenase, green indicates chlorosome and photochemical reaction center PscA, yellow indicates electrons being donated through oxidation reactions and photosynthesis, pink indicates reductant (NADPH) and chemical energy (ATP) produced during oxidation reactions and photosynthesis, and red indicates pathways of carbon assimilation through the reverse tricarboxylic acid cycle, as well as anaplerotic gluconeogenesis steps, using electrons derived from inorganic compound oxidation and energy derived from photosynthesis. The photosystem uses electrons derived from sulfide, hydrogen, thiosulfate, and iron oxidation and activates them using light energy, which allows proton pumping and ferredoxin reduction. Ferredoxin reduction is linked to photosystem activity, depicted by the double orange circle in electron. In the gray boxes, circles show whether marker genes are present or absent in the respective mOTUs in the legend. Genes are indicated by the following gene product abbreviations: PscA (photosystem I P700 chlorophyll *a* apoprotein A1), Sqr (sulfide-quinone oxidoreductase), Dsr (reverse dissimilatory sulfite reductase), Apr (adenylylsulfate reductase), Sat (sulfate adenylyltransferase), Sox (thiosulfohydrolase), Cyc2 (iron-oxidizing outer membrane *c*-type cytochrome), Hya (group 1d [NiFe]-hydrogenase), FNR (ferredoxin-NADP oxidoreductase), and Fd (ferredoxin).

oxidoreductases (Sqr) were encoded by all but two *Chlorobium* mOTUs (Fig. 2; Fig. S1B), whereas reverse dissimilatory sulfite reductases (i.e., DsrA) and flavocyto-chrome *c* sulfide dehydrogenase (FCC) were encoded by 71% and 53% of the mOTUs, respectively (Fig. 2). A more restricted trait was the capacity to oxidize thiosulfate via thiosulfohydrolase (i.e., SoxB). The corresponding gene was present in only 14% of the *Chlorobium* mOTUs, although it was found in all *Chlorobaculum* mOTUs (Fig. 2; Table S2). Others have suggested that members of the class *Chlorobia* have adapted to different environments by acquiring distinct electron transfer complexes through hori-zontal gene transfer (41–43). Even though genes for the Dsr and Sox complexes may

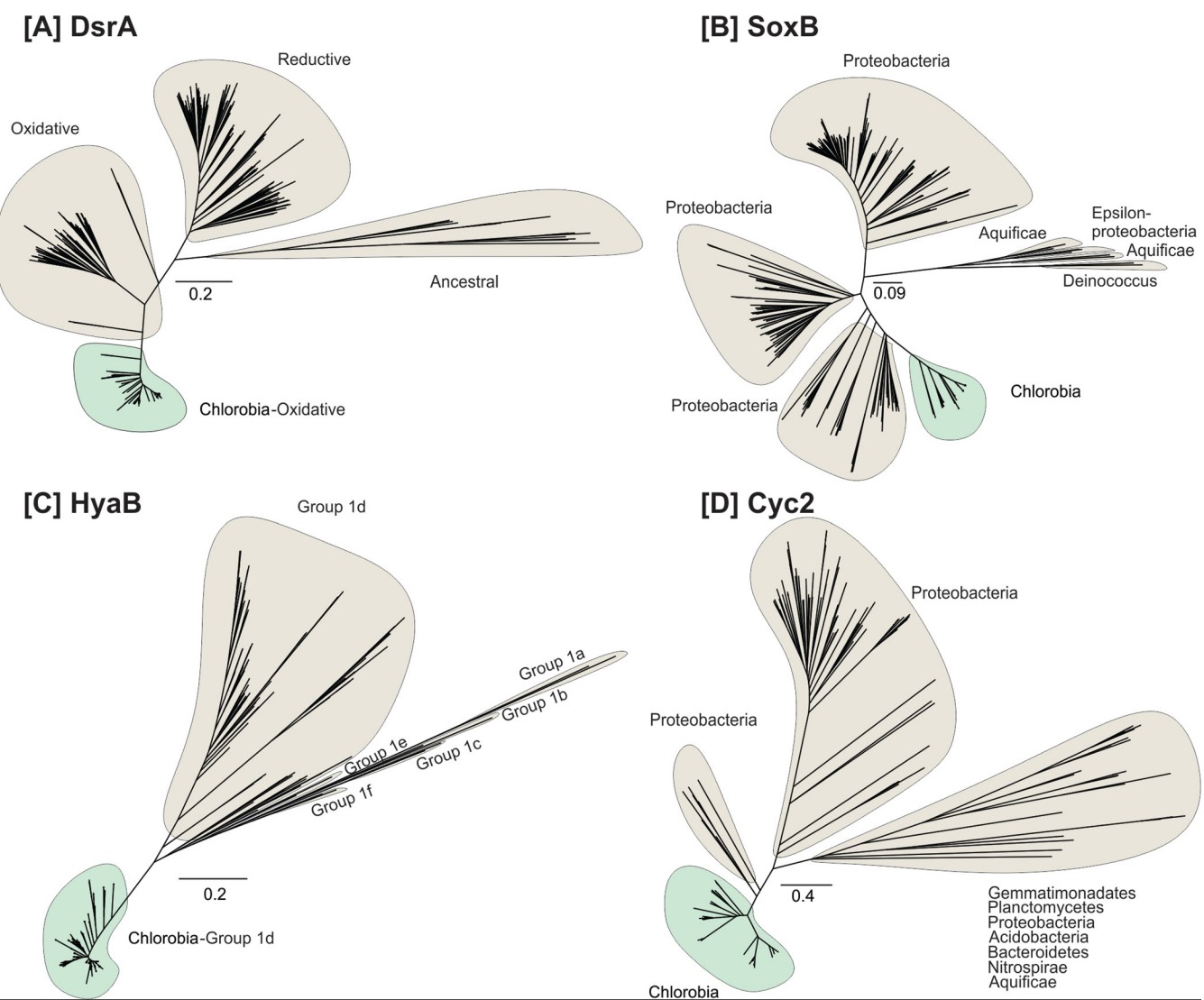

**FIG 6** Maximum likelihood phylogeny of the recovered protein sequences from reconstructed *Chlorobia*-affiliated MAGs/SAGs (highlighted in green) together with the reference protein sequences recovered from GenBank, with the exception of hydrogenases, for which sequences were collected from HydDB (highlighted in gray). Panels A to D show the phylogeny of the following protein sequences: (A) dissimilatory sulfite reductase (DsrA; $n = 551$), (B) thiosulfohydrolase (SoxB; $n = 294$), (C) group 1d [NiFe]-hydrogenase (HyaB; $n = 579$), and (D) putative iron-oxidizing cytochrome (Cyc2; $n = 306$).

have been horizontally acquired in this class (41), our phylogenetic trees show that genes encoding DsrA and SoxB form monophyletic clades, consistent with a model in which they were each acquired on one occasion during the evolution of *Chlorobia* and have potentially been lost from certain clades (Fig. 2 and Fig. 6A and B).

Genes encoding the catalytic subunits of group 1d and group 3b [NiFe]-hydrogenases (HyaB and HyhL, respectively) were detected in 82 and 91% of the *Chlorobium* mOTUs, respectively (Fig. 2 and 6C; Fig. S1C). The group 1d enzymes are known to support hydrogenotrophic respiration and anoxygenic photosynthesis in diverse bacterial lineages (44–46). It is likely that the electrons liberated by these enzymes are transferred to the photosynthetic reaction center to reduce ferredoxin and $NADP^+$ as a source of reductant for carbon fixation, nitrogen fixation, and other biosynthetic processes (47). Phylogenetic analysis confirmed that these subunits formed monophyletic radiations together with the hydrogenases retrieved from *Chlorobia* reference genomes (Fig. 6C). The physiological role of group 3b hydrogenases (HyhL) in *Chlorobia* genomes is less clear. These bidirectional cytosolic enzymes are likely to support either hydrogenotrophic carbon fixation or facultative hydrogenogenic fermentation in this class (Fig. S1D) (48, 49). Although uptake

hydrogenases have previously been reported for *Chlorobium* members (50), the ecological significance of H$_2$ metabolism has been overlooked in this class. Overall, the results suggest that H$_2$ metabolism is an ancestral and conserved trait of the class *Chlorobia*. This is also consistent with the widespread distribution of [NiFe]-hydrogenases across bacterial phyla (51, 52) and the conservation of group 1d lineages in sister classes within the phylum *Bacteroidota*.

Our study also shows that 40% of the *Chlorobium* mOTUs harbor *cyc2* gene homologs (Fig. 2 and 6D), encoding a potential outer membrane *c*-type cytochrome capable of ferrous iron oxidation (31, 53). This finding suggests that the distribution of the *cyc2* gene among *Chlorobium* genomes in lakes is higher than previously recognized. If the *cyc2* gene allows for extracellular electron transfer or ferrous iron oxidation, as speculated based on comparative genomics of cultured photoferrotrophs (31), this implies that *Chlorobia* members could play an important role in iron biogeochemistry in lake systems globally.

Metabolic flexibility has been found to be a key factor governing distributions of taxa across ecosystems with disturbances (54). However, we did not find any mOTU containing all the putative oxidation genes in the core genome (Fig. 2 and 5), nor did we find any correlation between how widespread an mOTU is and its capacity to use different electron donors. A broader sampling across different temporal and spatial scales could reveal whether metabolic versatility governs the prevalence and abundance of *Chlorobium* members on a global scale.

**Outlook of ecological roles of *Chlorobium* members.** At first glance, members of the *Chlorobia* appear to populate a very specific niche in water columns of global lakes and ponds. As we confirm in our data sets, they thrive under the oxycline in the anaerobic layer where light is still available. They have evolved physiological adaptations to low light intensities (4) and therefore are particularly well adapted to freshwater systems characterized by high concentrations of colored dissolved organic carbon (DOC) that absorb incident solar radiation (55). In these lakes, solar radiation only penetrates to shallow depths in the water column, constraining populations of *Cyanobacteria* and most other photoautotrophs to the surface layer (56). *Chlorobium* members may contribute up to 83% of the total annual productivity in such humic waters (25) and also seem to play an important role as diazotrophs (16, 17). Despite the specific niches occupied by *Chlorobium* populations, our work illustrates considerable metabolic versatility with regard to electron donors that may be used for carbon fixation. Through their variable capacity to recycle sulfide, hydrogen, and iron produced by dissimilatory sulfate reducers, hydrogenogenic fermenters, and dissimilatory iron reducers, *Chlorobium* members potentially influence the entire microbiome of anoxic freshwaters (57).

Our analyses describe the metabolic potential of environmentally relevant *Chlorobium* genomes, along with their geographical distributions, revealing both endemic and cosmopolitan clades and suggesting complex metabolic implications for those distributions. Distributions of *Chlorobia* populations appear governed by ecological factors beyond overall known metabolic potential. Moreover, our study identified clades of *Chlorobium* with abundant and cosmopolitan geographical distributions that do not show monophyletic clustering. Although comprehensive metadata and better geographical and temporal resolution would help identify the specific physicochemical mechanisms that lead to specific ecological advantages, our data nonetheless highlight previously overlooked ecological diversity for this globally distributed bacterial lineage.

## MATERIALS AND METHODS

**Collection of lakes and pond metagenomes.** We obtained 265 metagenomes from 42 locations that spanned subarctic to tropical regions as a part of a project aiming to study microbial diversity in anoxic freshwater environments (Fig. 1; Table S3) (29). Briefly, samples were collected between 2009 and 2018, with most obtained by biomass collection on 0.22-$\mu$m-pore Sterivex filters (Millipore). Oxygen concentrations were measured with a YSI 55 oxygen probe (Yellow Springs Instruments, Yellow Springs, OH, USA). For all samples, DNA was extracted using the PowerSoil DNA extraction kit (MoBio, Carlsbad, CA, USA) following the manufacturer's instructions. Sequencing libraries were prepared from 10 to 20 ng of purified DNA using the ThruPLEX DNA-seq Prep kit according to the manufacturer's preparation guide.

The quality of the libraries was evaluated using the Agilent Fragment Analyzer system and the DNF-910-kit. The adapter-ligated fragments were quantified by quantitative PCR (qPCR) using the Library quantification kit for Illumina (KAPA Biosystems/Roche) on a CFX384 Touch real-time PCR detection system (Bio-Rad) prior to cluster generation and sequencing. The sequencing libraries were pooled and subjected to cluster generation and paired-end sequencing with 150-base read-length S2/S4 flow cells and the NovaSeq6000 (Illumina Inc.) using the v1 chemistry according to the manufacturer's protocols. Sequencing was performed by the SNP&SEQ Technology platform (Uppsala, Sweden). Each sample generated an average of 10.5 Gb of sequence data. Moreover, we added six metagenomic libraries generated for a *Chlorobium* study in Canada, focusing on iron oxidation (31) and three additional metagenomic libraries as part of a 15-year time series analysis of Trout Bog, WI (30).

**Genome collection.** Assembling and binning the 265 metagenomes resulted in ~12,000 metagenome-assembled genomes (MAGs) from water bodies spanning five different countries (29). Of these, 454 belonged to the class *Chlorobia* and were used in this study. In brief, raw data were trimmed using Trimmomatic version 0.36 (58) and were then assembled with MEGAHIT version 1.1.13 (59) with default settings. Two assemblies were performed: single assemblies for all samples individually and 53 coassemblies that were performed mostly with lake-specific metagenome data (see Table S4 in the supplemental material). More detailed methods for sample collection, metadata, processing, metagenome generation, and analyses are reported elsewhere (29).

The relevant quality-controlled reads were mapped to all assemblies using BBMap (60) with default settings. Mapping results were used to bin contigs using MetaBAT version 2.12.1 (parameters –maxP 93 –minS 50 -m 1500 -s 10000) (61). Moreover, we collected genomes of the class *Chlorobia* from the GTDB and collected several other published MAGs that fulfilled the medium-quality threshold of ≥50% completeness and ≤5% contamination (62). In total, we compiled 509 genomes that comprised the following: 454 MAGs and 19 SAGs that were from stratified water bodies in Sweden, Finland, Canada, Switzerland, and Puerto Rico (29); 7 MAGs from enrichment culture or lake metagenomes of lakes at the International Institute for Sustainable Development Experimental Lakes Area (IISD-ELA; near Kenora, Canada) (31); 4 MAGs from Trout Bog, WI (30, 32); and 25 genomes (MAGs, SAGs, or isolates) from the GTDB (2, 63). Completeness and contamination were assessed with CheckM (64).

**Single-cell collection.** Cell sorting was done in 2016 on a MoFlo Astrios EQ sorter (Beckman Coulter, USA) using 488- and 532-nm lasers for excitation, a 70-$\mu$m nozzle, a sheath pressure of 60 lb/in$^2$, and 0.1-$\mu$m-pore-filtered 1$\times$ phosphate-buffered saline (PBS) as sheath fluid. An ND (neutral-density) filter (ND = 1) and masks M1 and M2 were used. The trigger channel was set to forward scatter (FSC) at a threshold of 0.025%, and sort regions were defined by autofluorescence using a 532-nm laser and band-pass filters 710/45 and 664/22. Sorted plates were stored at −80°C. Whole-genome amplification was conducted using the REPLI-g single-cell kit (Qiagen) following the manual's instruction, but with a reduced total reaction volume (12.5 $\mu$l). Amplified DNA was mixed thoroughly by pipetting up and down. After screening for bacterial 16S rRNA genes, DNA from confirmed *Chlorobia* members was sequenced on an Illumina HiSeqX v2.5 PE at 2 $\times$ 150 bp (29).

**mOTU analysis.** Average nucleotide identity (ANI) for all genome pairs was computed with FastANI version 1.3 (65), and the genomes were then clustered into metagenomic operational taxonomic units (mOTUs) with 70% completeness and 5% contamination thresholds. Genome pairs with ANI values above 95% were clustered into connected components. Additionally, less complete genomes (completeness above 50% yet below 70%) were recruited to the mOTU if the ANI similarity was above 95%. Classification of genomes was done with GTDB-Tk and sourmash version 1.0 (63, 66). The most complete MAG for each mOTU was selected as a representative, and a phylogenetic tree was calculated using GTDB-Tk version 102, with database release 89, using *Chloroherpetonaceae* as an outgroup (63). The aligned genome tree was loaded and curated in iTOL version 5.5.1 (67). Moreover, KO (KEGG Orthology) number functions were classified as core if the probabilities of their presence-absence profiles were higher under the assumption that they were in every genome, considering their incompleteness, than the converse probability (as computed by mOTUpan in mOTUlizer version 0.1.3 [https://github.com/moritzbuck/mOTUlizer]).

**Abundance profiles.** To explore abundance profiles, metagenomic libraries were subsampled to 1,000,000 reads. Metagenomic reads were then mapped to all collected *Chlorobia*-associated genomes (Table S1). Reads were mapped with 100% identity using BBMap (60). The results were then normalized to the relative abundance of each mOTU per metagenome. All relative abundances for all samples in each location were averaged. The cutoff for the presence of an mOTU was set to 0.03% per site (0.0003 in Fig. 3). Heat maps of abundance were calculated and plotted using the R packages ggplot2 and phyloseq with parameters NMDS (nonmetric multidimensional scaling) and Bray-Curtis for choosing the order of the x axes (68–71). Depth-discrete abundance profiles were plotted using R packages ggplot2, phyloseq, and cowplot (69–71).

**Metabolic genes.** The metabolic potentials of the *Chlorobia* MAGs, SAGs, and reference genomes were reconstructed based on eggNOG-mapper (72, 73) annotations. Ten of the metabolic marker gene products—PsaA (photosystem I P700 chlorophyll *a* apoprotein A1), AclB (ATP-citrate lyase beta-subunit), NifH (nitrogenase iron protein), FCC (flavocytochrome *c* sulfide dehydrogenase), Sqr (sulfide-quinone oxidoreductase), DsrA (dissimilatory sulfite reductase A subunit), SoxB (thiosulfohydrolase), Cyc2 (outer membrane monoheme *c*-type cytochrome), HyaB (group 1d [NiFe]-hydrogenase large subunit), and HyhL (group 3b [NiFe]-hydrogenase large subunit)—were further annotated in reconstructed MAGs, SAGs, and reference genomes using Diamond version 0.9.31 (74) against custom-built databases for each marker protein with an 80% query coverage threshold. A 50% identity threshold was used for all marker gene products, except for an 80% threshold used for the PsaA protein. Annotations were further validated by constructing phylogenetic trees.

**Single-gene phylogeny.** Maximum likelihood trees were constructed using the amino acid sequences for six metabolic marker proteins: PsaA (photosystem I P700 chlorophyll *a* apoprotein A1), NifH (nitrogenase iron protein), Sqr (sulfide-quinone oxidoreductase), DsrA (sulfite reductase A subunit), SoxB (thiosulfohydrolase), Cyc2 (outer membrane monoheme *c*-type cytochrome), HyaB/HyaA (group 1d [NiFe]-hydrogenase large subunit and small subunit), and HyhL (group 3b [NiFe]-hydrogenase large subunit). Reference genes for each marker were collected from NCBI GenBank, with the exception of hydrogenases, for which sequences were collected from HydDB (46). For each metabolic marker protein, sequences retrieved from the *Chlorobia* mOTUs were aligned against reference protein sequences using ClustalW in MEGA7 (75, 76). Evolutionary relationships were visualized by constructing a maximum likelihood phylogenetic tree. Specifically, initial trees for the heuristic search were obtained automatically by applying Neighbor-Join and BioNJ algorithms to a matrix of pairwise distances estimated using a JTT model and then selecting the topology with superior log likelihood value. All residues were used, and trees were bootstrapped with 50 replicates.

**Data availability.** All data sets are available in public repositories under NCBI project accession numbers PRJEB38681, PRJNA518727, and PRJNA534305. Accession numbers of the genomes are available (Table S1), as are the accession numbers of metagenomes (Table S3). Assembling and binning of the original data set used scripts available at https://github.com/moritzbuck/metasssnake, and general processing scripts for this project are available at https://github.com/moritzbuck/0026_Chlorobi.

## SUPPLEMENTAL MATERIAL

Supplemental material is available online only.

**FIG S1**, JPG file, 2.5 MB.
**TABLE S1**, CSV file, 0.1 MB.
**TABLE S2**, CSV file, 0.02 MB.
**TABLE S3**, CSV file, 0.03 MB.
**TABLE S4**, CSV file, 0.03 MB.

## ACKNOWLEDGMENTS

The work was primarily funded by Science for Life Laboratory, the Olsson-Borgh, Knut and Alice Wallenberg Foundations (grant KAW 2013.0091), and Kungl. Vetenskapsakademiens stiftelser (BS2018-0108). K.D.M. acknowledges funding from the United States National Science Foundation Microbial Observatories Program (MCB-0702395), the Long-Term Ecological Research Program (NTL-LTER DEB-1440297), and an INSPIRE award (DEB-1344254). J.D.N. acknowledges Discovery Grant and Strategic Partnership Grant for Projects funding from the National Sciences and Engineering Research Council of Canada (NSERC). C.G. is supported by a National Health & Medical Research Council EL2 Fellowship (APP1178715). S.B. acknowledges funding from the Swedish Research Council and the Swedish Research Council Formas. The authors acknowledge additional support and resources from the National Genomics Infrastructure in Stockholm funded by the Science for Life Laboratory, the Swedish Research Council, and SNIC/Uppsala Multidisciplinary Center for Advanced Computational Science for assistance with massively parallel sequencing and access to the UPPMAX computational infrastructure (project SNIC2020/5-19). The funders had no role in study design, data collection and interpretation, or the decision to submit the work for publication.

We offer our gratitude to the Cree and Inuit communities in Whapmagoostui-Kuujjuarapik for giving us access to their ancestral lands.

S.L.G. and S.P. conceptualized the research idea. All authors provided data. M.B., M.M., S.L.G., J.M.T., C.G., and K.D.M. performed data analysis. S.L.G., M.M., C.G., and S.P. did literature searches. S.L.G., M.M., C.G., and S.P. drafted the manuscript, and all authors contributed to writing and editing the manuscript.

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
