## [Reviewer comments · mSystems]

Freshwater *Chlorobia* exhibit metabolic specialization among cosmopolitan and endemic populations

Sarahi Garcia, Maliheh Mehrshad, Moritz Buck, Jackson T suji, Josh Neufeld, Katherine McMahon, Stefan Bertilsson, Chris Greening, and Sari Peura

Corresponding Author(s): Sarahi Garcia, Stockholm University

Review Timeline:

Submission Date:	November 13, 2020
Editorial Decision:	January 15, 2021
Revision Received:	February 17, 2021
Accepted:	April 9, 2021

Editor: Matthias Hess

Reviewer(s): Disclosure of reviewer identity is with reference to reviewer comments included in decision letter(s). The following individuals involved in review of your submission have agreed to reveal their identity: Rachel Poretsky (Reviewer #1); Jakob Pernthaler (Reviewer #2)

Transaction Report:

DOI: <https://doi.org/10.1128/mSystems.01196-20>

January 15, 2021

Dr. Sarahi L Garcia
Stockholm University
Department of Ecology, Environment and Plant Sciences
Stockholm
Sweden

Re: mSystems01196-20 (Freshwater Chlorobia exhibit metabolic specialization among cosmopolitan and endemic populations)

Dear Dr. Sarahi L Garcia:

Below you will find the comments of the reviewers.

To submit your modified manuscript, log onto the eJP submission site at <https://msystems.msubmit.net/cgi-bin/main.plex>. If you cannot remember your password, click the "Can't remember your password?" link and follow the instructions on the screen. Go to Author Tasks and click the appropriate manuscript title to begin the resubmission process. The information that you entered when you first submitted the paper will be displayed. Please update the information as necessary. Provide (1) point-by-point responses to the issues raised by the reviewers as file type "Response to Reviewers," not in your cover letter, and (2) a PDF file that indicates the changes from the original submission (by highlighting or underlining the changes) as file type "Marked Up Manuscript - For Review Only."

Due to the SARS-CoV-2 pandemic, our typical 60 day deadline for revisions will not be applied. I hope that you will be able to submit a revised manuscript soon, but want to reassure you that the journal will be flexible in terms of timing, particularly if experimental revisions are needed. When you are ready to resubmit, please know that our staff and Editors are working remotely and handling submissions without delay. If you do not wish to modify the manuscript and prefer to submit it to another journal, please notify me of your decision immediately so that the manuscript may be formally withdrawn from consideration by mSystems.

Sincerely,

Matthias Hess

Editor, mSystems

Journals Department
Reviewer comments:

Reviewer #1 (Comments for the Author):

This is a really nice, comprehensive study that uses many different metagenomic, SAG, and genome datasets to compare Chlorobia populations. It is apparent that the MG and SAG datasets were taken as part of a larger study, so a lot of the details are not presented in this manuscript. It would be really helpful, though, to add some of this information here, specifically how they were sequenced, how big the datasets were, read lengths, and how well each metagenome covered the total community community. This would be useful for interpreting the results. Related to this, I think it would be good to know the relative abundances of Chlorobia at each location. I'm not sure if it would change anything, but the results are only for abundant Chlorobia and only for samples in which Chlorobia make up a large enough fraction of the community to be assembled.

Specific comments:

Line 128-129: What does it mean to be composed exclusively of genomes from isolates? Are they identical genomes?

Line 141: what are the read lengths?

Section beginning line 177: This data was not presented here, correct? I suggest either showing the results or not discussing it (I prefer the former.)

Section beginning line 201: I'm curious about gene variants. Some of these are highly conserved, but others might show phylogenetic relationships associated with niche. I would like to see this addressed if possible.

Line 253: Why would more samples be likely to reveal correlations? There were a lot of samples presented here.

Methods: please provide some more information about the metagenomes. I understand it's presented elsewhere, so a summary would suffice here.

Line 291-2 Please explain what was co-assembled.

Figure 1 (and 3): Please use different symbols and colors for the sample locations; it is so hard to

discern the orange from the red, especially with the same shapes.

Fig 2: Line 612-2: Change "where" to "were" (two times)

Fig 2: I can't see any of the gene data. It seems like it is cut off from the figure.

Reviewer #2 (Comments for the Author):

This is an extensive report about the biogeography of members of the Genus *Chlorobium*, as reconstructed from metagenomic information. It was found that most members of the genus, including all widely spread or abundant "species" (termed mOTUs) were from uncultivated lineages. The observed distribution patterns could, however, not be related to "metabolic flexibility", i.e., the most widely spread mOTUs were physiologically no more versatile than others.

While this study may not be the most sparkling example of hypothesis-driven research, it is as solid as a neutron star with respect to data collection and evaluation. Also, despite the state-of-the-art bioinformatic analysis, it is a refreshingly old-school type of manuscript: It recounts the familiar story that isolates are not representative of natural communities, but does so at the level of a single conspicuous genus of freshwater bacteria. This is novel and interesting, in particular since the authors also include aspects of functional ecology: Cultured *Chlorobium* species are not only phylogenetically distinct from the mOTUs in freshwater systems, they also differ in their metabolic capacities.

Specific & minor comments:

The text of the figure legend violates manuscript style requirements in that it does not have proper line spacing. Such things make the job of a reviewer harder. Please don't make a habit of this.

The results and discussion section includes too much technical information about the analysis (e.g., l.119-121, l.141-142, l.190-192) that should be moved to the methods section.

The paragraphs l.119-134 and l.136-147 are somewhat confusing and should be re-written or re-organized. Right now, it sounds to me like the numbers don't add up: in l. 128 it is reported that 57 mOTUs were of the genus *Chlorobium*, of which 12 were isolates. Yet in l. 144 there are only 42 *Chlorobium* mOTUs left. And, to make things totally weird, there are actually 43 mOTUs affiliated with the genus *Chlorobium* in Fig. 2 (yes, I am pedantic, I counted several times)

The text to Fig. 2 first left me mystified. Where IS all that information mentioned in the legend? Then I realized that it was probably cut off during pdf conversion. And so was Fig 5! Unfortunately, this error by itself will make another round of review necessary. A strong argument for checking the generated pdf prior to submission!

Fig. 3: The relative abundance classes in this figure are ... challenging. Please tame your graphics program. Also, I do not understand which dimension they are, since the lower cutoff stated in the legend (0.03%) is much higher than the 3 lower abundance classes.

l.183-185: do not confuse proportion within the metagenome with absolute abundances. Only total cell numbers could tell if these bacteria actually became more (or if others became less). This also casts doubt on your speculation of seasonality.

l.252: this is an interesting finding and should be elaborated in more detail. It is no surprise that your findings disagree with those of a study in "frequently disturbed habitats" (ref 55). Also, would you please check if the genome sizes of the cosmopolitan mOTUs were different from the others?

l.271: which ecological factors do you have in mind?

Supplementary Table "Supplemental_Material02" seems to make the interesting claim that some samples were collected in 2021, 2022, and 2025

Minor comments:

L. 152-153: This statement is self-evident and could be removed

l.152, 153: "we found" used twice in consecutive sentences

Fig 1: Switzerland?

Fig. 2 legend: l. 612 "where assembled"?

Fig 4 legend: l. 642:"and while"; l.645,647: make up your mind to use the abbreviation or not.

l.270, 271: distributions that distribute?

Reviewer #1 (Comments for the Author):

This is a really nice, comprehensive study that uses many different metagenomic, SAG, and genome datasets to compare Chlorobia populations. It is apparent that the MG and SAG datasets were taken as part of a larger study, so a lot of the details are not presented in this manuscript. It would be really helpful, though, to add some of this information here, specifically how they were sequenced, how big the datasets were, read lengths, and how well each metagenome covered the total community. This would be useful for interpreting the results. Related to this, I think it would be good to know the relative abundances of Chlorobia at each location. I'm not sure if it would change anything, but the results are only for abundant Chlorobia and only for samples in which Chlorobia make up a large enough fraction of the community to be assembled.

Thank you for your comments. We have now added information on how the samples were collected and processed prior to sequencing in the methods section. We also added information about read lengths and the average size of the metagenomic datasets. Moreover, we added details to Table S3 about the size (in number of bases) of each sample.

Figure 4 shows the relative abundances of all *Chlorobia* present in each location. The most abundant mOTUs have their own colors, and we also included information about the mOTUs that are clustered under “others” in the figure legend.

Specific comments:

Line 128-129: What does it mean to be composed exclusively of genomes from isolates? Are they identical genomes?

This means that 13 mOTUs only included genomes from isolates. These genomes were non identical. We added the following wording to make this clear.

“Among 57 Chlorobium-associated mOTUs, 13 were composed exclusively of **different genomes from previously described **non-identical** isolates (2), including Chlorobium (Chl.) phaeobacteroides, Chl. limicola, Chl. luteolum, Chl. ferrooxidans, and Chl. phaeoclathratiforme (Figure 2 **and Table S2**).”**

Line 141: what are the read lengths?

Read libraries are paired end with length of 150 bp. We added this information in the methods section as we revised the manuscript in response to an earlier comment of this reviewer.

Section beginning line 177: This data was not presented here, correct? I suggest either showing the results or not discussing it (I prefer the former.)

It seems the paragraph was not clear enough and now we have reworded this passage to clarify that the data are presented in Figure 4B

“Several mOTUs appear to be endemic to specific geographical and seasonal niches. For example, we observed that mOTUs 22, 17, and 13 were temporally stable within Lake Lomtjärnen (Figure 4B). These mOTUs were observed across four timepoints:.”

Section beginning line 201: I'm curious about gene variants. Some of these are highly conserved, but others might show phylogenetic relationships associated with niche. I would like to see this addressed if possible.

We agree, and this is what we did with several genes that we considered most relevant to the niche selection and expansion of Chlorobia representatives in lake ecosystems. We have hence reconstructed detailed phylogeny of the genes DsrA, SoxB, HyaB, Cyc2, PsaA, Sqr, NifH, and HyhL presented in the Figure 6 and Figure S1. Additionally, in Figure 2 we show the distribution of these genes in different mOTUs and their affiliation to either the core- or accessory genome. In the present work we have focused on the known genes involved in electron transfer as we believe these genes and their distribution pattern hold great potential for defining the niche in Chlorobia.

Line 253: Why would more samples be likely to reveal correlations? There were a lot of samples presented here.

We revised this sentence for clarity:

“A broader sampling across different temporal and spatial scales could reveal whether metabolic versatility governs the prevalence and abundance of Chlorobium members on a global scale.”

Methods: please provide some more information about the metagenomes. I understand it's presented elsewhere, so a summary would suffice here.

We have added a method summary, as described above.

Line 291-2 Please explain what was co-assembled.

We have added a new Table S4 that provides details about samples that were co-assembled.

Figure 1 (and 3): Please use different symbols and colors for the sample locations; it is so hard to discern the orange from the red, especially with the same shapes.

We changed the USA samples to purple for clarity in both Figures 1 and 3.

Fig 2: Line 612-2: Change "where" to "were" (two times)

This change was made as suggested.

Fig 2: I can't see any of the gene data. It seems like it is cut off from the figure.

We appreciate that the reviewer noticed this error. We confirm that the merged document contains complete figures this time.

Reviewer #2 (Comments for the Author):

This is an extensive report about the biogeography of members of the Genus *Chlorobium*, as reconstructed from metagenomic information. It was found that most members of the genus, including all widely spread or abundant "species" (termed mOTUs) were from uncultivated lineages. The observed distribution patterns could, however, not be related to "metabolic flexibility", i.e., the most widely spread mOTUs were physiologically no more versatile than others.

While this study may not be the most sparkling example of hypothesis-driven research, it is as solid as a neutron star with respect to data collection and evaluation. Also, despite the state-of-the-art bioinformatic analysis, it is a refreshingly old-school type of manuscript: It recounts the familiar story that isolates are not representative of natural communities, but does so at the level of a single conspicuous genus of freshwater bacteria. This is novel and interesting, in particular since the authors also include aspects of functional ecology: Cultured *Chlorobium* species are not only phylogenetically distinct from the mOTUs in freshwater systems, they also differ in their metabolic capacities.

We appreciate the comprehensive review of our work.

Specific & minor comments:

The text of the figure legend violates manuscript style requirements in that it does not have proper line spacing. Such things make the job of a reviewer harder. Please don't make a habit of this.

The results and discussion section includes too much technical information about the analysis (e.g., l.119-121, l.141-142, l.190-192) that should be moved to the methods section.

Although we retain some information for the reader about the core analyses performed, we have reduced excessive technical detail from the results and discussion section. We believe that several concepts, such as mOTUs, are important to be introduced together with the results for the manuscript to be accessible and understandable for readers.

The paragraphs l.119-134 and l.136-147 are somewhat confusing and should be re-written or re-organized. Right now, it sounds to me like the numbers don't add up: in l. 128 it is reported that 57 mOTUs were of the genus *Chlorobium*, of which 12 were isolates. Yet in l. 144 there are only 42 *Chlorobium* mOTUs left. And, to make things totally weird, there are actually 43 mOTUs affiliated with the genus *Chlorobium* in Fig. 2 (yes, I am pedantic, I counted several times)

In the paragraph that starts in line 119 we talk about general statistics of our mOTUs. As we wrote in in the beginning of the paragraph, there are 71 *Chlorobia* mOTUs. Of those, 57 are *Chlorobium* and 13 of these are isolates. Thanks to this reviewer comment, we discovered one isolate that was not counted previously – this is corrected now.

In the paragraph starting at line 149, we discuss abundant mOTUs with respect to reads mapped to them. Of the 71 mOTUs in our study, 45 mOTUs mapped reads above the 0.0003 cutoff and, of those, 42 were classified as *Chlorobium*.

Finally, in Figure 2 we only show 53 (not 43) of the 57 *Chlorobium* mOTUs. We are not showing 4 isolates and we marked those in Table S2. We now also added a sentence to the legend of Figure 2 to reflect this change.

Together, the numbers are now all correctly stated in the manuscript.

The text to Fig. 2 first left me mystified. Where IS all that information mentioned in the legend? Then I realized that it was probably cut off during pdf conversion. And so was Fig 5! Unfortunately, this error by itself will make another round of review necessary. A strong argument for checking the generated pdf prior to submission!

We apologize - all figures are shown properly for the re-submission.

Fig. 3: The relative abundance classes in this figure are ... challenging. Please tame your graphics program. Also, I do not understand which dimension they are, since the lower cutoff stated in the legend (0.03%) is much higher than the 3 lower abundance classes.

0.03% relative abundance is the same as the 0.0003 that is shown in the figures. We have added the number of samples per lake/pond to Figure 3 and added a note about the % value to make this clear.

l.183-185: do not confuse proportion within the metagenome with absolute abundances. Only total cell numbers could tell if these bacteria actually became more (or if others became less). This also casts doubt on your speculation of seasonality. Add to the text something on cell counts

In order to fully address this concern, we have added a few sentences to make our point more clear for future readers:

“In addition, the relative abundance of these mOTUs changed according to the sampling time. For example, mOTU13 increased in relative abundance for the September sampling point. We predict that some endemic mOTUs may occupy specific seasonal niches and suggest that a more temporally resolved sampling effort, together with cell counts to calculate absolute abundances, would help test this hypothesis.”

I.252: this is an interesting finding and should be elaborated in more detail. It is no surprise that your findings disagree with those of a study in "frequently disturbed habitats" (ref 55). Also, would you please check if the genome sizes of the cosmopolitan mOTUs were different from the others?

We realize the sentences might have not been clear enough, so we re-arranged the text to make our point more clear. We also checked genome sizes of cosmopolitan mOTUs and found no significant differences as compared to the other genomes.

The text now reads as follows:

“Metabolic flexibility has been found to be a key factor governing taxa distributions across ecosystems with disturbances (54). However, we did not find any mOTU encoding all the putative oxidation genes in the core genome (Figure 2 and 5), nor did we find any correlation between how widespread an mOTU is and their capacity to use different electron donors. A broader sampling across different temporal and spatial scales could reveal whether metabolic versatility governs the prevalence and abundance of Chlorobium members on a global scale.”

I.271: which ecological factors do you have in mind?

We have changed the sentence and it now reads as follows: “Distributions of Chlorobia populations appear governed by ecological factors beyond overall metabolic potential.”

Supplementary Table "Supplemental_Material03" seems to make the interesting claim that some samples were collected in 2021, 2022, and 2025

Thank you for catching this formatting error – the supplemental table now includes the full month and year format for all samples and this will now be clear for future readers.

Minor comments:

L. 152-153: This statement is self-evident and could be removed

We removed this sentence as suggested.

l.152, 153: "we found" used twice in consecutive sentences

We removed one sentence as suggested.

Fig 1: Switzerland?

Thank you for spotting this typographic error.

Fig. 2 legend: l. 612 "where assembled"?

Changed where to were.

Fig 4 legend: l. 642:"and while"; l.645,647: make up your mind to use the abbreviation or not.

We used the abbreviation.

l.270, 271: distributions that distribute?

We changed the sentence to the following:

“Distributions of Chlorobia populations appear governed by ecological factors beyond overall metabolic potential.”

April 9, 2021

Dr. Sarahi L Garcia
Stockholm University
Department of Ecology, Environment and Plant Sciences
Stockholm
Sweden

Re: mSystems01196-20R1 (Freshwater Chlorobia exhibit metabolic specialization among cosmopolitan and endemic populations)

Dear Dr. Sarahi L Garcia:

Your manuscript has been accepted, and I am forwarding it to the ASM Journals Department for publication. For your reference, ASM Journals' address is given below. Before it can be scheduled for publication, your manuscript will be checked by the mSystems senior production editor, Ellie Ghatineh, to make sure that all elements meet the technical requirements for publication. She will contact you if anything needs to be revised before copyediting and production can begin. Otherwise, you will be notified when your proofs are ready to be viewed.

- Minimum resolution of 1280 x 720
- .mov or .mp4. video format
- Provide video in the highest quality possible, but do not exceed 1080p
- Provide a still/profile picture that is 640 (w) x 720 (h) max

We recognize that the video files can become quite large, and so to avoid quality loss ASM suggests sending the video file via <https://www.wetransfer.com/>. When you have a final version of

the video and the still ready to share, please send it to Ellie Ghatineh at eghatineh@asmusa.org.

Sincerely,

Matthias Hess
Editor, mSystems

Journals Department
Supplemental Material: Accept
Supplemental Material: Accept
Supplemental Material: Accept
Supplemental Material: Accept
Supplemental Material: Accept